# *Meta*Biome: a multiscale model integrating agent-based and metabolic networks to reveal spatial regulation in gut mucosal microbial communities

Javad Aminian-Dehkordi,[1] Andrew Dickson,[1] Amin Valiei,[1] Mohammad R. K. Mofrad[1,2]

**ABSTRACT**  Mucosal microbial communities (MMCs) are complex ecosystems near the mucosal layers of the gut essential for maintaining health and modulating disease states. Despite advances in high-throughput omics technologies, current methodologies struggle to capture the dynamic metabolic interactions and spatiotemporal variations within MMCs. In this work, we present *Meta*Biome, a multiscale model integrating agent-based modeling (ABM), finite volume methods, and constraint-based models to explore the metabolic interactions within these communities. Integrating ABM allows for the detailed representation of individual microbial agents each governed by rules that dictate cell growth, division, and interactions with their surroundings. Through a layered approach—encompassing microenvironmental conditions, agent information, and metabolic pathways—we simulated different communities to showcase the potential of the model. Using our *in-silico* platform, we explored the dynamics and spatiotemporal patterns of MMCs in the proximal small intestine and the cecum, simulating the physiological conditions of the two gut regions. Our findings revealed how specific microbes adapt their metabolic processes based on substrate availability and local environmental conditions, shedding light on spatial metabolite regulation and informing targeted therapies for localized gut diseases. *Meta*Biome provides a detailed representation of microbial agents and their interactions, surpassing the limitations of traditional grid-based systems. This work marks a significant advancement in microbial ecology, as it offers new insights into predicting and analyzing microbial communities.

**IMPORTANCE**  Our study presents a novel multiscale model that combines agent-based modeling, finite volume methods, and genome-scale metabolic models to simulate the complex dynamics of mucosal microbial communities in the gut. This integrated approach allows us to capture spatial and temporal variations in microbial interactions and metabolism that are difficult to study experimentally. Key findings from our model include the following: (i) prediction of metabolic cross-feeding and spatial organization in multi-species communities, (ii) insights into how oxygen gradients and nutrient availability shape community composition in different gut regions, and (iii) identification of spatiallyregulated metabolic pathways and enzymes in *E. coli*. We believe this work represents a significant advance in computational modeling of microbial communities and provides new insights into the spatial regulation of gut microbiome metabolism. The multiscale modeling approach we have developed could be broadly applicable for studying other complex microbial ecosystems.

**KEYWORDS**  gut microbiome, mucosal microbial community, metabolic interactions, spatiotemporal features, cross-feeding, spatial regulation

Address correspondence to Mohammad R. K. Mofrad, mofrad@berkeley.edu.

Javad Aminian-Dehkordi and Andrew Dickson contributed equally to this article. The author order was determined alphabetically.

M.R.K.M. is the co-founder of Nexilico, Inc., a start-up developing AI-driven microbiome engineering technologies.

The human gut microbiota exhibits significant spatial and temporal diversities, with varying microbial populations observed across its cross-section and length (1–3). This complex microbiome consisting of a wide range of cell types and taxa plays a critical role in food metabolism and is vital for maintaining intestinal mucosal integrity, establishing homeostasis, and supporting overall health (4–6). Understanding these spatiotemporal characteristics is key for grasping how microbiome composition and functional features change dynamically in response to environmental factors (7, 8). These changes can, in turn, affect host functions: environmental shifts can induce a microbiome composition richer in harmful phenotypes, which potentially undermine host immunity and may disrupt the integrity of the gut barrier (9). On the contrary, the adaptability of microbial communities to changing environments can lead to adaptive shifts in microbiome composition, microbial interactions, and overall function (10).

Gut microbes are hypothesized to perform complex functions by forming intricate microbial communities (11). Experimental studies revealed that a significant portion of these microbes inhabit densely packed structures on food particles and along the gut mucosa (12–14), hereafter referred to as mucosal microbial communities (MMCs). The functional characteristics of MMCs suggest potential roles in gut metabolism and pathogenesis, particularly because of their stability and persistence, which confer microbiome resilience, even after antimicrobial interventions (12, 15). Despite the recognized importance of MMCs, there remains a profound gap in our understanding of their structural and functional dynamics, largely attributable to the limitations inherent in current research methodologies.

The complexity of microbial ecosystems typically necessitates sophisticated computational metabolic models to predict their behavior, systematically explore hypothetical scenarios, and even design microbial communities with desired traits (16, 17). Despite significant strides in this domain (18, 19), capturing the full spectrum of microbial dynamics requires overcoming several computational and conceptual hurdles. Metabolic modeling of microbial communities, particularly through genome-scale metabolic models (GEMs), offers a window into their metabolic interactions (20). Extending traditional flux balance analysis (FBA) through a multi-compartment approach allows for more detailed modeling of these interactions (21), optimizing community-level objective functions that often emphasize biomass production. However, this approach tends to oversimplify the diverse objectives and interactions among community members. In response, methodologies like OptCom (22) and MICOM (23) integrated multi-level optimization and network properties to capture the dual objectives of individual microbes and the community at large.

A pivotal yet often disregarded aspect in genome-scale metabolic modeling is accurately representing biomass reaction fluxes and microbial growth rates (24, 25). Traditional algorithms falter by not enforcing synchronized growth rates across community members, often resulting in unrealistic dominance of faster-growing species. This discrepancy stems from a mismatch between specific substrate utilization rates and metabolite exchange fluxes within the community, leading to skewed predictions of community dynamics. The necessity of enforcing a uniform growth rate arises from the need to more accurately reflect natural microbial interactions and ensure balanced community dynamics. This limitation prompted the development of cFBA (26) and its improved version, SteadyCom (27), noted for their adeptness at differentiating between the community growth as a collective behavior and the relative abundances of individual microbes.

Another limitation of FBA-based methods is their inability to directly account for spatial distributions and temporal dynamics, which has motivated their integration with agent-based models (ABMs). Using this approach, the metabolite exchange fluxes obtained by FBA dynamically adjust metabolite concentrations within the environment, subsequently predicting the impacts on bacterial populations. This modeling paradigm shifts the focus from aggregate, population-level analyses to the behaviors and interactions of individual microbes modeled as discrete agents (28). ABMs excel in

capturing the spatial and temporal dynamics of microbial communities, which enables the exploration of individual microbes in terms of growth, competition, and collaboration within a shared environment (29–32). This approach effectively addresses the complexities of biomass reaction fluxes and differential growth rates across a community.

Previous ABM-based models integrated with metabolic networks have significantly advanced our understanding of microbial communities in general, yet they have generally lacked a specific focus on interactions within MMCs and the underlying mechanical behavior between species. Most of these previous models primarily focused on population-level interactions, often without the detailed spatiotemporal resolution required to effectively study biofilm structure. For instance, BacArena (33), an R package based on MatNet (34), adopts an ABM approach that places each cell on a grid block, which represents microbial communities in a spatial layout. While this setup allows for species interactions and spatial positioning, BacArena is limited by its cell-as-grid-block design. It fails to capture important biological phenomena like agent cell size, movement dynamics, or intracellular variability, and it restricts the ability to model dynamic changes in transport properties and diffusion mechanisms with high fidelity. COMETS, implemented in Java with MATLAB and Python toolboxes, goes a step further by leveraging dynamic FBA and adopts a population-centric approach over ABM, focusing solely on calculating the average metabolic activities of cell populations within each grid unit (35). As a result, it is less suited for studying heterogeneous environments, particularly MMCs, where local nutrient gradients and cell-level interactions are crucial. Versluis et al. (36) also used a similar approach to study microbial transitions in the infant gut microbiota, which, while informative, lacked the spatial resolution needed to address MMC-specific dynamics. Recently, MiMICS has developed a multiscale model using ABM and incorporating spatially resolved transcriptomic data to identify distinct metabolic states within *Pseudomonas aeruginosa* biofilms, providing insights into the development of targeted strategies to manage biofilm-associated infections (37).

We develop *Meta*Biome, a multiscale model of MMCs, to jointly account for fine-grained metabolic interactions within these communities. We show that our multiscale model offers unprecedented granularity and high flexibility, enabling a detailed system analysis. Previously, by integrating ABM and finite volume methods (FVMs), we simulated MMCs through conceptually driven, metabolism-based interactions, and, in conformity with other investigations (38–40), we showed that competition, neutralism, mutualism, and cooperation govern MMC dynamics (41, 42). Findings indicated that mutualistic relationships provide significant fitness advantages to microbial communities in the gut. Specifically, cross-feeding patterns promote interspersion and structural homogeneity, which enhances the integrity of the microbial community. These patterns are particularly effective in dampening compositional variations and establishing stable environments for mutualistic newcomers. This can contribute to the stability and functionality of the gut microbiota.

In the present study, we present *Meta*Biome and extend our previous model by incorporating GEMs into the framework. Our model comprises three distinct layers of information essential to modeling MMCs: the environment layer that contains MMC microbes, extracellular space, and concentration gradients with FVMs used to solve the transport equations for metabolites. The agent layer retains positional information and models the behaviors and interactions of agents. It models their metabolism through nutrient consumption, metabolite production, and processes of growth and division. Intracellular fluxes are computed using GEMs, while metabolites diffuse across the domain in response to their gradients. The dynamic interactions between agents and their environment across defined temporal and spatial scales enable *Meta*Biome to reveal unique spatiotemporal effects and emergent properties of MMCs.

*Meta*Biome offers a distinct framework for simulating mucosal microbial dynamics and their interactions mediated by metabolites by advancing beyond traditional FBA-based approaches to integrate environmental diversity and spatial heterogeneity—

essential for modeling MMCs in complex environments like the gut. It leverages a shoving algorithm that allows agents to physically interact, capturing growth patterns within confined spaces, while a wall constraint module enhances spatial accuracy by capturing boundary interactions. *Meta*Biome also includes a transport layer that models concentration gradients across spatial regions, providing a detailed view of microbial behavior in varied environments. Furthermore, it supports the dynamic transition of microbial states between planktonic and attached states, a capability that mirrors microbial population shifts. Implemented in Python with a high-performance NumPy backend, *Meta*Biome combines computational efficiency with accessibility, enabling an in-depth simulation of large-scale MMCs and revealing insights into microbial responses to environmental changes that traditional dynamic models cannot capture (for more details on the differences of our platform to previous approaches, please see Table S1 and Fig. S1 in the Supplementary Information file).

We posit that our multiscale model can introduce perturbations across varying resolutions by linking cellular characteristics at the genome level to phenotypic properties at the community level. It can also be smoothly tuned to predict the effect of parametric changes through computational iterations, offering a substantial reduction in time and cost compared to experimental studies. The predictive capability of this framework can equip researchers with crucial insights into the role of key gut microbial metabolites, positioning the model as a valuable tool in systems pharmacology and the development of targeted therapies.

## RESULTS

### Model evaluations

Briefly, our multiscale model utilizes a 2D domain to simulate the MMCs within the gut. Each lattice cell on the grid accommodates multiple metabolites and bacterial populations. Metabolite concentrations are updated to account for diffusion, microbial

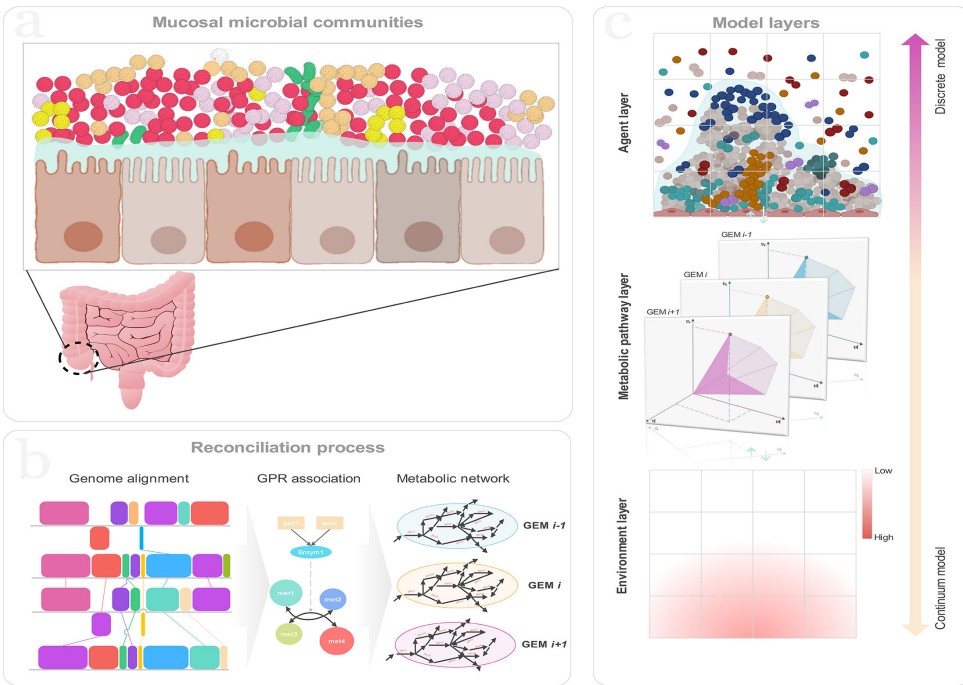

**FIG 1** An overview of the simulation process: (a) mucosal microbial communities are complex assemblages of microorganisms inhabiting mucosal surfaces, playing critical roles in host physiological processes; (b) schematic representation of the reconciliation process used for the reconstruction of genome-scale metabolic models used in this study; and (c) the multiscale model includes three layers, namely, the agent, metabolic pathway, and environment layers.

consumption, and production over time. To initialize the model, microbial agents are randomly populated across the domain, resulting in initial attachment to the substratum following a short-term free planktonic movement. This results in a random seeding layer that supports the growth of the bacterial communities. Nutrients diffuse into the domain from the upper boundary, where they are consumed by the bacteria. As the simulation progresses, bacterial populations metabolize, grow, and divide. At the same time, metabolites are continuously diffused, consumed, and produced (for detailed model description, including assumptions, equations, and boundary conditions, please refer to Methods). This dynamic process captures the spatial and temporal distributions of the attached microbial population and metabolite concentrations, which offers insights into the development and interactions of microbial communities under different environmental conditions (Fig. 1).

To evaluate the capacity of our model in predicting metabolic interactions and microbial abundance, we conducted simulations using different scenarios. First, we modeled different microbial communities with established cross-feeding patterns from the literature. These included simplified representations of the human gut microbiota colonized in germ-free mice, focusing on butyrate production's relevance to gut health (Scenario I). This cross-feeding pattern was selected, as butyrate production is a well-documented metabolic function crucial to gut homeostasis. This allows us to assess the model's predictive capacity for short-chain fatty acid (SCFA) cross-feeding and capture microbial interactions involving butyrate production, which has broader implications for gut health. We also simulated an *in vitro* microbial community with a specific focus on SCFA production (Scenario II). Then, our multiscale model was applied to a gut microbial community with a high-protein diet.

We began the simulations with a simplistic two-species community for SCFA production, which expanded to include the contribution of another microbe to the community. We simulated another microbial community with four species to further assess the performance of our model in predicting metabolic interactions and microbial abundance. For each scenario, we compared the simulation results with available experimental data by employing relevant GEMs and involving relevant metabolites. The characteristics of the GEMs used in the simulations are detailed in Table S2. All the GEMs underwent a reconciliation process (see Methods for more details) and were compared with their available counterparts on the Virtual Metabolic Human database (28, 43) (Fig. S2).

In the next step, we used our multiscale model to predict the impact of a high-protein diet averaged from the dietary patterns of 45 overweight individuals on the metabolic interactions of MMCs in the gut (Table S9). We simulated MMCs at two different regions within the gut, namely, the proximal small intestine and the cecum, and imposed oxygen gradients to mimic the physiological conditions present within the two gut regions. To simulate the diet, the nutrients of the high-protein diet were introduced to the domain from the upper surface (Table S10). The nutrient profile for the high-protein diet did not include fatty acids, focusing on amino acids, carbohydrates, and dietary fiber to capture protein-driven SCFA production and cross-feeding patterns. The products, including SCFAs, could be either absorbed by endothelial cells from the bottom or consumed by the other microbes. The results were compared with experimental data from mucosal samples. Fecal samples were also used to compare our results, as they provided the most accessible source of microbiota data available. We performed different analyses of the bacteria at both the fluxomic and community levels to investigate underlying metabolic interactions, identify potential enzymes that may play critical roles, and warrant further studies.

### Scenario I. SCFA production by E. rectale and B. thetaiotaomicron

We first simulated the interactions of *Eubacterium rectale* and *Bacteroides thetaiotaomicron*—representative species of the phyla *Firmicutes* and *Bacteroides*, respectively—mediated by nutrients and fermentation products. By employing the reconstructed

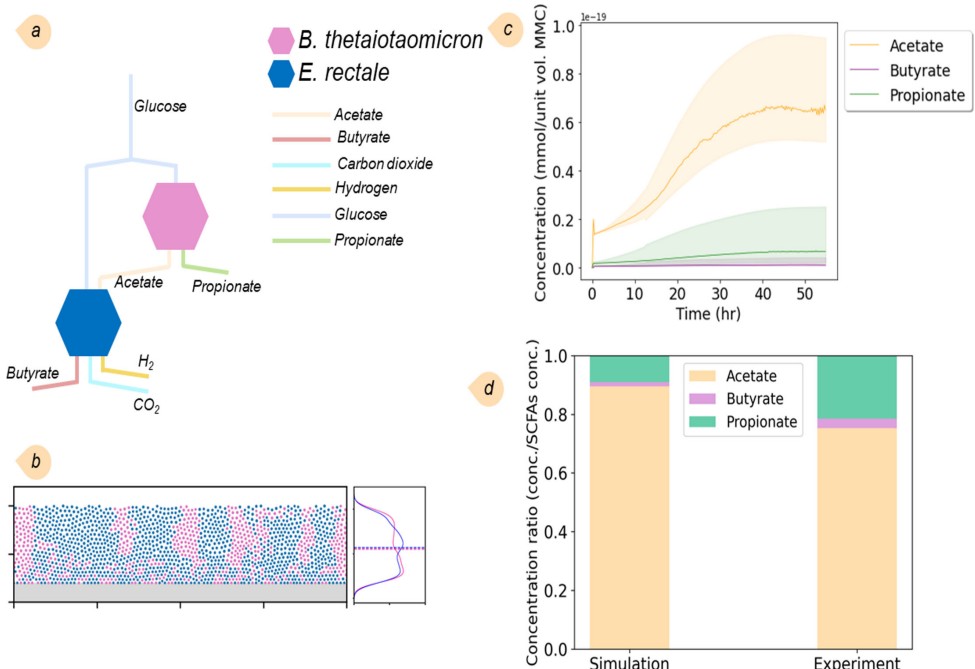

**FIG 2** Results of the simulation of the two-species MMC for SCFA production. (a) Sketch of metabolic interactions of *B. thetaiotaomicron* and *E. rectale* where butyrate is produced by *E. rectale*. (b) Spatial orientation and KDE of the community. (c) Concentrations of three main SCFAs, including acetate, propionate, and butyrate, are produced by the bacteria in the community. (d) Comparison of predicted SCFA levels obtained at steady state with SCFA concentrations in the feces of colonized germ-free mice (the experimental data obtained from reference 44). The sum of SCFA concentrations was used as the reference for normalization.

GEMs, we assessed the capability of our multiscale model to predict SCFA production and compared the results with metabolite profiles previously available from mono-colonized germ-free mice experiments. Simulations showed *B. thetaiotaomicron* consumes glucose to produce acetate and propionate, and *E. rectale* feeds on glucose in addition to acetate produced by *B. thetaiotaomicron* to produce metabolites, including butyrate (see Fig. 2a for the metabolic interaction network). Bacterial interaction through these metabolic interactions results in lateral and longitudinal growth and the formation of distinct clusters for each species (see population density profiles in Fig. 2b). We previously showed that elevated bacterial cross-feeding through metabolic products induces the formation of interconnected regions within bacterial biofilms, fostering intermixing among constituent populations (41). Here, bacterial species show little tendency to intermix with each other, as *E. rectale* relies on both carbohydrates as its primary nutritional source in addition to acetate as a metabolic product. Nonetheless, both species are able to occupy a substantial niche space featuring a steady-state condition for populations of bacteria and chemical species, as shown in Fig. 2c for SCFA (data are not shown for metabolites, including succinate, carbon dioxide, and hydrogen). The concentrations of acetate, propionate, and butyrate at these conditions align closely with the experimental data showing substantial amounts of acetate and small amounts of propionate and butyrate (Fig. 2d).

Next, we included *Methanobrevibacter smithii*, a well-known gut methanogen, alongside the two previously studied species. Despite its relatively low abundance within the gut microbiota, *M. smithii* plays a crucial role in the digestive process. Our interest in simulating this three-species community stemmed from its capability to convert carbon dioxide and hydrogen into methane, which significantly influences community dynamics. The simulation provides a simplified depiction of this community's interaction network (Fig. 3a). *B. thetaiotaomicron* consumes the predominant portion

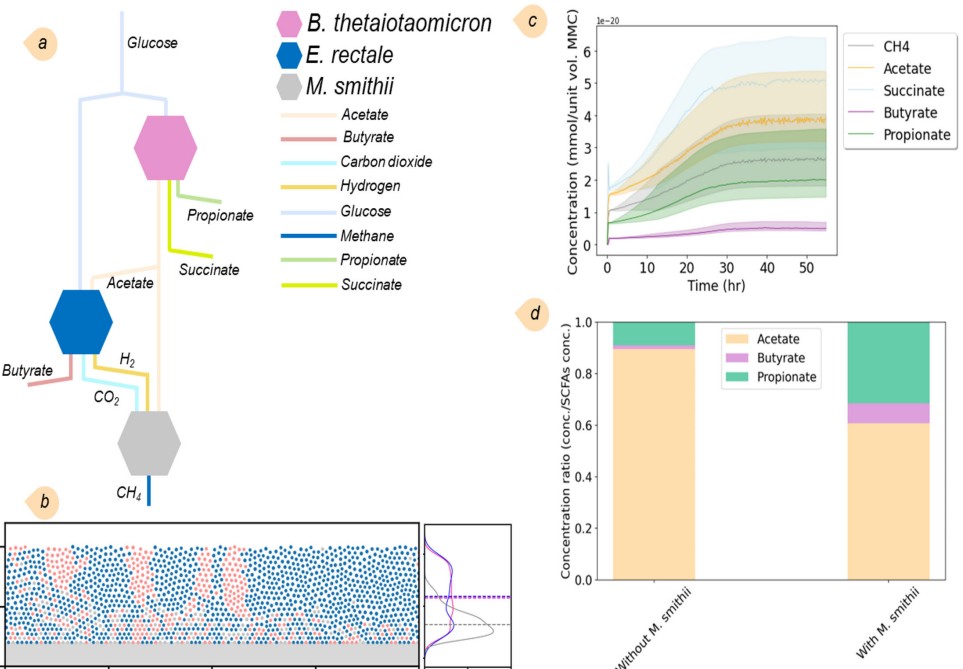

FIG 3 Results of the simulation of the three-species MMC for SCFAs and methane production: (a) sketch of the metabolic interactions of *B. thetaiotaomicron*, *E. rectale*, and *M. smithii*, where butyrate and methane are produced by *E. rectale* and *M. smithii*, respectively; (b) spatial orientation and KDE of the community; *M. smithii* bacteria tend to lie near the bottom surface; (c) concentrations of SCFAs, including acetate, propionate, butyrate, and succinate, as well as methane, produced by the bacteria in the community; and (d) comparison of the predicted SCFA levels obtained in this community at steady state with SCFA concentrations with those obtained in Fig. 2. The sum of the SCFA concentrations was used as the reference for normalization.

of the glucan, yielding SCFAs, such as acetate, propionate, and succinate. *E. rectale* subsequently utilizes a fraction of the acetate, along with glucan, to produce butyrate and carbon dioxide. *M. smithii*, in turn, assimilates a portion of acetate and carbon dioxide to synthesize methane. Fig. 3b, showing the spatial orientation and kernel density estimation (KDE) analysis of the community structure, reveals that, unlike *B. thetaiotaomicron* and *E. rectale*, *M. smithii* avoided clustering and dispersed laterally across the domain. This dispersion is attributed to *M. smithii*'s significantly lower biomass flux and slower replication rate than the other two species. As a result, *M. smithii* is unable to form clusters because the faster-growing bacteria shove it away and fill voids. Fig. 3c illustrates the changes in SCFA concentrations over time until a steady state is reached. We compared the SCFA levels in this community with those recorded in the previous two-species community. The results show a reduction in acetate levels while propionate levels increase, subsequently elevating the concentration of butyrate (Fig. 3d).

### Scenario II. Metabolic interactions of a four-species community

In another scenario, we studied the metabolic interactions of a microbial community consisting of *E. rectale*, *B. thetaiotaomicron*, *Bifidobacterium adolescentis*, and *R. bromii* (Fig. 4a). Based on the *in vitro* experimental data, we assumed that the bacteria utilize cellobiose and starch as carbohydrate sources for growth. Relative abundances of species in the community were compared to those obtained in *in vitro* experiments. While the results for *B. thetaiotaomicron* and *E. rectale* were in agreement, deviations were noted for the other two species (Fig. S3). The simulation results confirmed the production of SCFAs and amino acids by the community. Figure 4b represents the abundance and biomass distribution of *B. thetaiotaomicron* averaged across the thickness. Before time $t_2$, voids persist within the layer, permitting bacterial proliferation until the entire domain

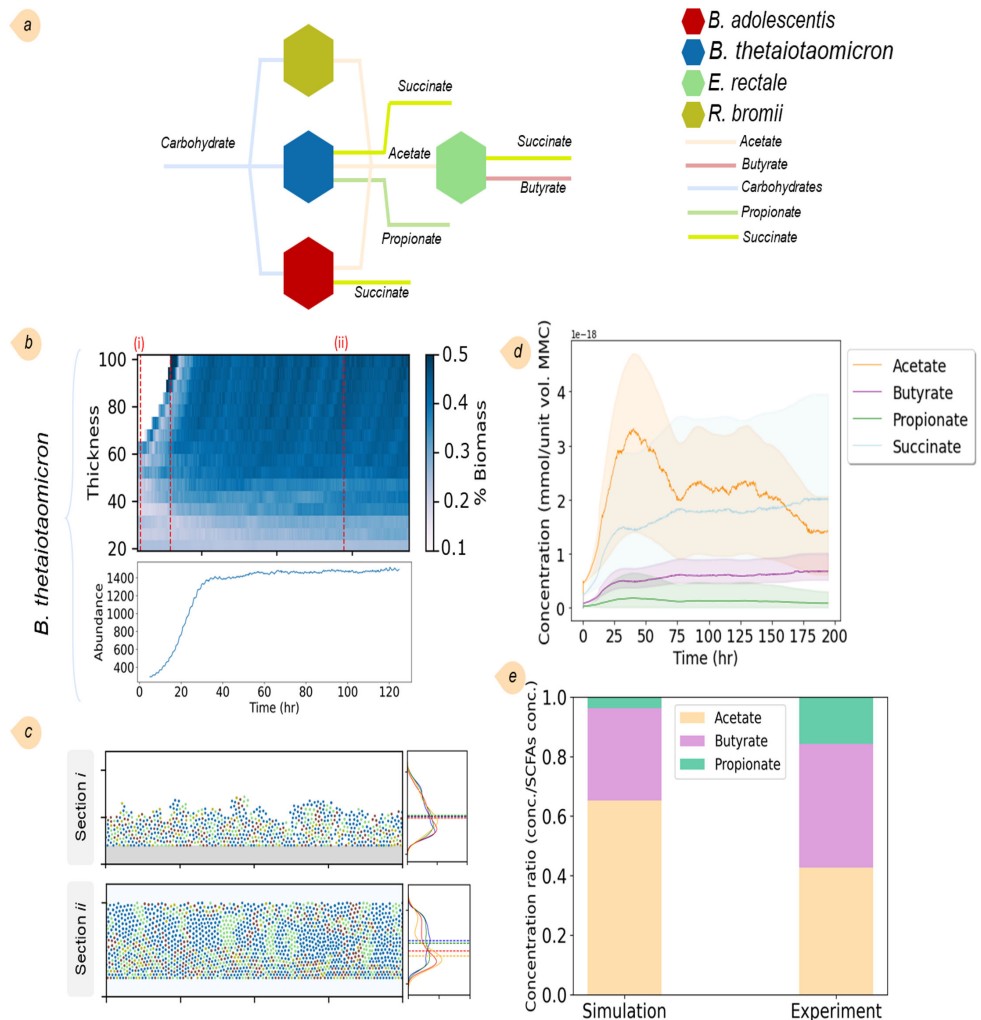

**FIG 4** Results of the simulation of the four-species community comprising *E. rectale*, *B. thetaiotaomicron*, *B. adolescentis*, and *R. bromii*: (a) sketch of the metabolic interactions of the community where different metabolites are produced by the bacteria; (b) biomass distribution of *B. thetaiotaomicron* averaged across the thickness and its abundance across the whole domain over time; (c) the ecosystem evolution and the spatial orientations and KDEs of the community at two different time points; (d) concentrations of SCFAs, including acetate, propionate, butyrate, and succinate, produced by the bacteria in the community; and (e) comparison of the predicted SCFAs levels obtained after the steady state in the simulations with SCFA concentrations resulting from community culture experiments performed in M2 medium supplemented with starch and cellobiose (the experimental data obtained from reference 45). The sum of the SCFA concentrations was used as the reference for normalization.

is occupied by bacteria of all species at $t_2$. Subsequently, as bacteria continue to grow and duplicate, they shove each other away, displacing and expelling those proximate to the upper boundary. Thereafter, both the concentrations of metabolites and the overall biomass exhibited minimal fluctuations, remaining nearly constant. Figure 4c, depicting the spatial orientation and KDE analysis of the community structure, affirms that *E. rectale* and *B. thetaiotaomicron* can form clusters in this community. Meanwhile, *B. adolescentis* only forms clusters of relatively small sizes, and *R. bromii* fails to form any clusters due to its substantially lower bacterial abundance.

The metabolic interaction network within the community indicates that acetate production is facilitated by *B. thetaiotaomicron*, *B. adolescentis*, and *R. bromii*, among which *B. thetaiotaomicron* is the predominant producer (Fig. 4a). Concurrently, *E. rectale* plays a role in acetate consumption. *E. rectale* also contributed to this ecosystem by

secreting butyrate, whereas *B. thetaiotaomicron* is solely responsible for propionate production. While acetate and propionate concentrations obtained from the simulations parallel the observed trend in experimental data, the predicted butyrate levels appear to be underestimated (Fig. 4e). The capacity to produce succinate is shared among *E. rectale*, *B. thetaiotaomicron*, and *B. adolescentis*, highlighting the multifaceted roles these microorganisms play in SCFA production in this microbial ecosystem. The simulations further elucidated the extent of bacteria contributions to amino acid production. Although the concentration of amino acids was lower compared to SCFAs, our analysis showed that *E. rectale* and *B. thetaiotaomicron* have the highest part in amino acid production, accounting for approximately 72% of the amino acids synthesized (Fig. S4).

## Case study: MMCs in different regions of the gut

The heterogeneity of the gut microbiota composition exhibits significant variability along the longitudinal axis of the gastrointestinal tract, such as different sections of the intestine. Current understanding of the mechanisms responsible for this diversity is limited and underexplored. Considering the profound sensitivity of microbial interactions to the ambient oxygen levels (46), we modeled MMCs at two distinct gastrointestinal niches—the proximal small intestine and the cecum—differing in oxygen gradients and nutrient levels. In the proximal small intestine, the concentration of oxygen in the lumen (~10 mm HG) is considerably lower than that within the intestinal wall (~59 mm HG) and at the villus tip (~22 mm Hg). This indicates a significant gradient from the vascular region to the luminal region. Similarly, there is an oxygen gradient with higher levels observed intravascularly in the colon, about 42 mm Hg in the submucosa, and much lower levels close to the crypt–villi interface (5–10 mm Hg). The oxygen level in the sigmoid colon lumen is estimated to be about 3 mm Hg (47). Both regions illustrate a consistent pattern of higher intravascular oxygen levels compared to those in the luminal areas. It is noted that some studies also reported a negligible (virtually nil) oxygen presence detected in the lumen (48).

Using our multiscale approach, a microbial community model with six key bacteria present within the human gut was constructed. The community includes one species in *Actinobacteria* (*B. adolescentis*), one in *Bacteroidetes* (*B. thetaiotaomicron*), three in *Firmicutes* (*E. rectale*, *Faecalibacterium prausnitzii*, and *Limosilactobacillus reuteri*), and one in *Proteobacteria* (*Escherichia coli*). These species were reported to be the most dominant bacteria in overweight individuals (45). The properties of corresponding GEMs are delineated in Table S2.

We first included all the exchange reactions common to at least two organisms, totaling to 230 metabolites (Table S11). After performing preliminary experiments, we accounted for 70 metabolites to precisely compute concentration gradients in the environment layer (Table S12). Any model with any of these exchange reactions had the potential to secrete or uptake the metabolites as a result of parsimonious FBA (*p*FBA) simulations run for each agent. The simulations were initiated by setting an averaged high-protein diet, including carbohydrates and amino acids (Datasheet DS2).

The simulations on the cecal region yielded results in agreement with the experimental data regarding species abundance derived from mucosal (48) and fecal samples (45) (Fig. S5). Our results showed that within this community, *E. coli* and *B. adolescentis*, known for their aerotolerant properties (49, 50), demonstrated increased biomass concentrations in the proximal small intestine compared to the cecum model. *E. rectale*, identified as the predominant bacteria in this community, exhibited minimal variation in biomass across the examined regions. A similar pattern was observed for *B. thetaiotaomicron*. Meanwhile, *L. reuteri* showcased a modest increase in biomass within the cecum, whereas *F. prausnitzii* showed reduced biomass in the proximal small intestine model. Furthermore, we compared changes in the relative abundances across different phyla within the gut microbiota based on simulations from the proximal small intestine to the cecum. The comparison is supported by data from mucosal samples analyzed via

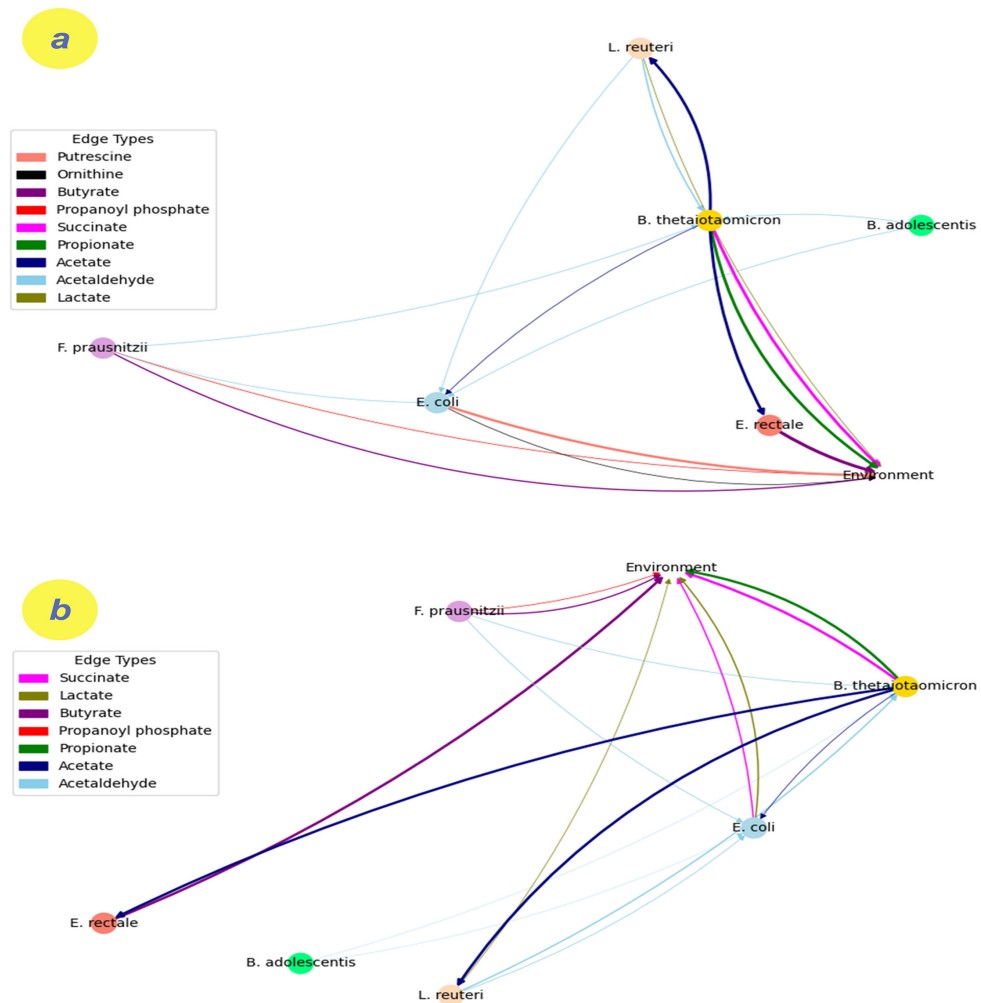

**FIG 5** Potential cross-feeding interactions within the community through evaluating average metabolic uptake and excretion fluxes across different species for different regions: (a) the proximal intestine and (b) the cecum.

16S rRNA sequencing (48), which aligns with the experimental data. This agreement in bacterial abundance suggests that the model can provide insights into the differences in the community within these distinct gastrointestinal regions.

We studied the potential cross-feeding interactions of the community in the proximal small intestine and the cecum by analyzing the average metabolic uptake and secretion fluxes of different species in the community (Fig. 5; Fig. S6). This analysis highlights the results for metabolites with concentrations exceeding 0.1, as these have been identified to significantly influence the metabolic interaction within the community. To enhance our comprehension of the metabolites directly released into the domain, an additional node termed *'environment'* was incorporated into the metabolic network graph (see Fig. 5). To help understand different interactions, amino acids were excluded from the graphs. The graph suggests that in both scenarios, SCFAs, especially acetate, are the main cross-feeding products, with *B. thetaiotaomicron* emerging as a "super cross-feeder" by producing acetate, succinate, and propionate. *E. coli* is another key node by assimilating acetate produced by *B. thetaiotaomicron* and acetaldehyde produced by *F. prausnitzii*, *L. reuteri*, and *B. adolescentis. E. coli,* interestingly, was able to synthesize putrescine and ornithine in the proximal intestine, which is not seen in the cecum model (for more details, see the metabolism of *E. coli* for both scenarios in the next section). Acetaldehyde, a common terminus of biochemical pathways in microbial communities, acts as an oxidant capable of interacting with other biomolecules under specific conditions

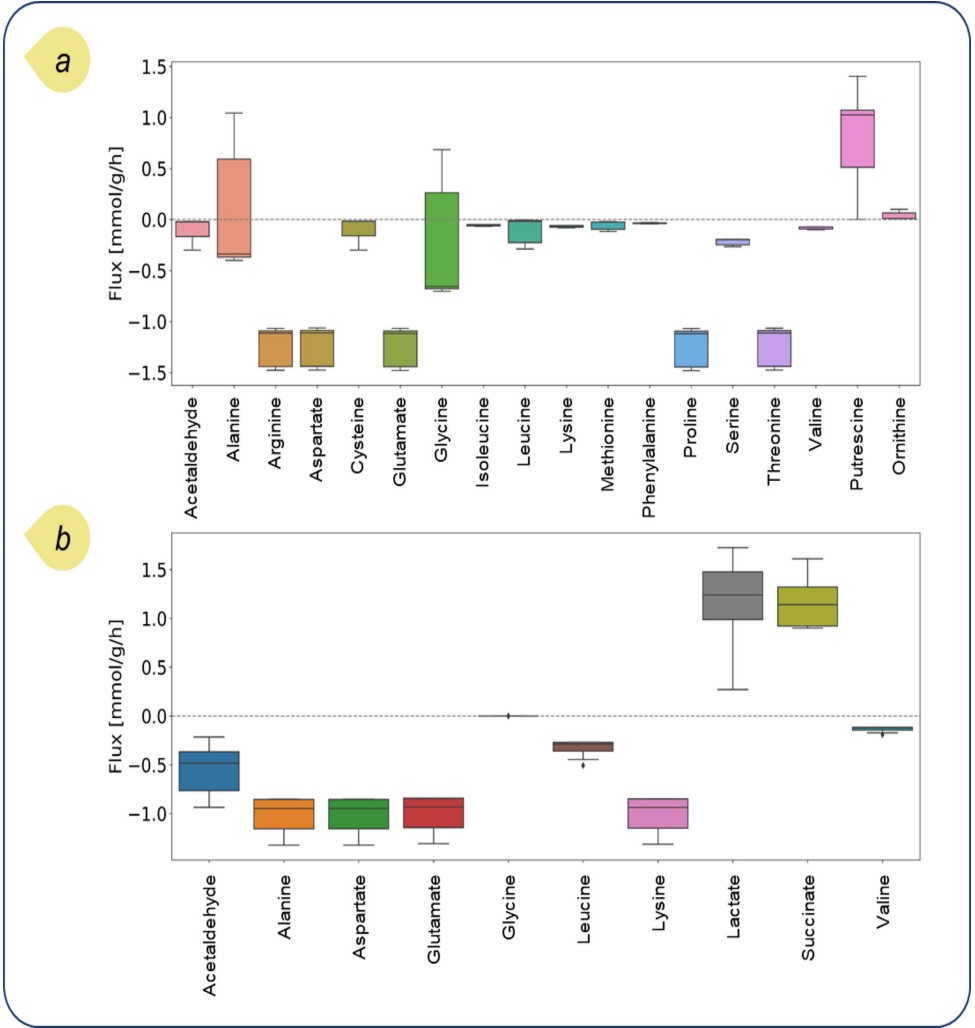

**FIG 6** Differences in the distribution of exchange fluxes related to all *E. coli* bacteria across microbiota in (a) the proximal intestine and (b) the cecum. The exchange fluxes were optimized based on the pFBA algorithm.

(51). The consumption of acetaldehyde hints at the possibility that the gut microbiota can regulate endogenous acetaldehyde systematically (52), although this remains a speculative hypothesis pending further investigation. Within this community, *E. rectale* stands as the primary butyrate producer in both scenarios. While in the proximal intestine, *L. reuteri* is the main lactate producer, in the cecum, *E. coli* also contributes to lactate production.

We studied the distribution of optimized exchange fluxes of bacteria across microbiota within the domain, inhabiting distinct regions of the gut, particularly the proximal intestine and the cecum. The results are presented in Fig. S7 (for *E. coli*, refer to Fig. 6). The observed heterogeneity in metabolic fluxes for certain compounds suggests that potential metabolic flexibility within individual microbes enables them to adeptly modulate their metabolic processes in response to the availability of substrates. *B. thetaiotaomicron*, for instance, exhibited relatively uniform fluxes, with rates above $-1$ mmol/g$_{DW}$/h for most amino acids, yet demonstrated markedly variable fluxes for SCFAs, including acetate, succinate, and propionate, across the domain. While *B. adolescentis* had minimal variations in amino acid fluxes, consuming only alanine, glutamate, and lysine, other species showed a marked difference by utilizing a diverse array of amino acids. Focusing on essential amino acids in the simulation, we observed that the importance of arginine is in that it was consumed by all the bacterial phyla,

except for *B. adolescentis*, which was in alignment with experimental data reported before (53).

In this section, we concentrate on *E. coli* as a model organism, given the copious experimental data available concerning its metabolism. This wealth of empirical evidence enables a comparison of our findings from the *in-silico* model. As stated before, the proximal intestine model predicts the biosynthesis of putrescine and ornithine, both pivotal in maintaining gut homeostasis (54). Reports confirm that the presence of arginine within the gut can lead to the synthesis of putrescine and, in turn, ornithine in different species, including *E. coli* (55), which is the case for our proximal intestine model. Regarding the *E. coli* metabolic network, arginine can be converted to agmatine to produce putrescine and urea using agmatinase, which is important for this metabolic pathway. Putrescine can be converted non-oxidatively under 2-oxoglutarate aminotransferase, involving the transfer of an amino group from putrescine to 2-oxoglutarate and producing 4-aminobutanal and glutamate. This aminotransferase, as a part of the arginine catabolism pathway, prefers aliphatic diamines, such as putrescine, which is smaller and less hydrophobic, over other substrates like ornithine and gamma-aminobutyric acid (56). Given this mechanism, the presence of putrescine and ornithine might be less noticeable in the cecum due to the efficient conversion of putrescine to other compounds. Thus, this might be a possible reason why we do not see putrescine and ornithine produced in the cecum model (57).

Higher levels of ornithine can lead to the production of agmatine, a biogenic amine, which has recently gained importance as a potential therapeutic agent for depression treatment (58, 59). On the contrary, reduced levels of ornithine may increase the risk of autism spectrum disorder (60). Dysregulation in ornithine and arginine metabolism has been associated with several disorders. The interplay between these metabolites is complex and can affect conditions ranging from pulmonary arterial hypertension to probably autism spectrum disorders (61).

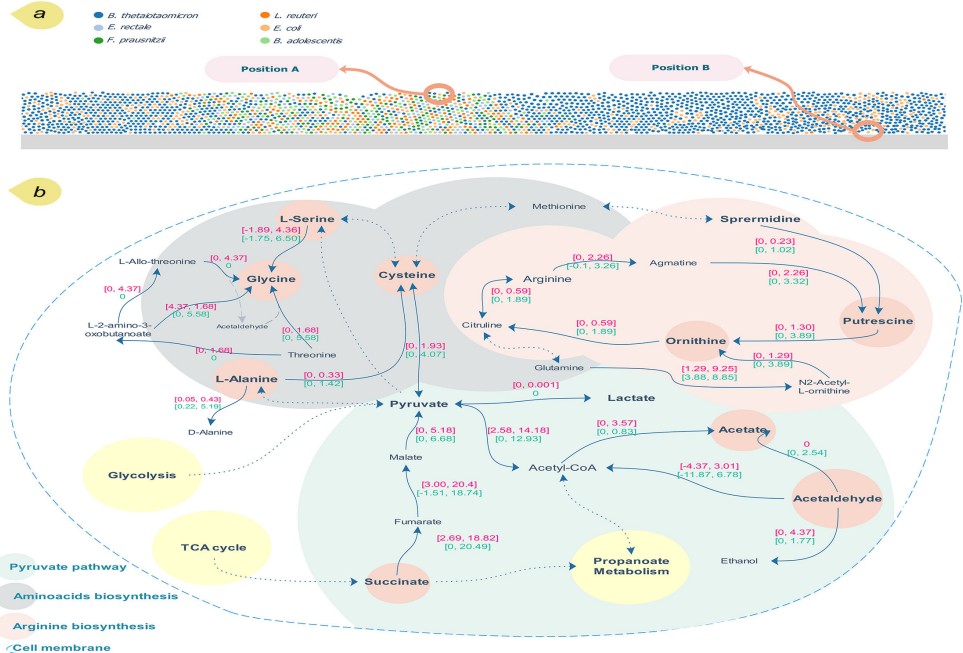

**FIG 7** Results of spatial regulation of reaction fluxes for *E. coli* in the proximal small intestine scenario, (a) simulated environment showing the spatial position of two *E. coli* bacteria: one located near the top boundary with lower access to oxygen but higher access to lumen-derived metabolites (position A), and the other, near the bottom boundary, closer to the host surface (position B). (b) Suboptimal FVA simulations for *E. coli*. Red and green values correspond to agents at positions A and B, respectively, highlighting the differences in flux patterns due to spatial positioning.

Comparative analysis of metabolic fluxes among various individual *E. coli* bacteria in the proximal small intestine suggests spatial heterogeneity in alanine and glycine metabolism. The flux distribution of alanine and glycine across the entire *E. coli* populations in the proximal intestine varies from [−0.45, 1.05] and [−0.85, 0.55] mmol/g$_{DW}$/h, respectively, indicating distinct metabolic profiles. In contrast, these values shift to [−1.25, −0.9] and 0 in the cecum, respectively.

To elucidate the localized growth dynamics of alanine and glycine, we selected two distinct *E. coli* agents at different lattice cells within the domain, position A at [362, 79] and position B at [734, 3] with different extracellular metabolite concentrations (Fig. 7). One strain exhibited uptake of both alanine and glycine, while the other was characterized by the secretion of these amino acids. Notably, we did not observe any strain exhibiting opposing fluxes, that is, all strains either consumed or produced both alanine and glycine. Next, we used previously optimized metabolic constraints to run suboptimal flux variability analysis (FVA). FVA facilitates the modulation of the objective function within a permissible range, in our case, ≥90% of the maximal biomass production rate.

Regarding Fig. 7, differences in some fluxes are evident, which indicates the spatial regulation of related enzymatic reactions influenced by spatial gradients of carbon and nitrogen sources, as well as oxygen levels. Alanine racemase, for instance, catalyzes the interconversion between L- and D-alanine, a process crucial for peptidoglycan biosynthesis and cell wall integrity, which makes the enzyme a significant antibiotic target. It also regulates the amino acid pool by balancing L- and D-alanine levels. The FVA results highlight alanine racemase's role in L-alanine secretion likely influenced by oxygen levels and availability of nutrients. Interestingly, Díaz-Pascual et al. (62) revealed spatial regulation of alanine within *E. coli* colonies, where secretion occurs in anoxic regions rich in glucose and ammonia. They reported that exogenous alanine can also be used as a nitrogen source or weak carbon source under oxic conditions.

To expand our investigation, we performed suboptimal FVA across all the *E. coli* bacteria within the domain, applying the pre-optimized constraints. This approach was designed to systematically identify potential enzymes and their corresponding genes

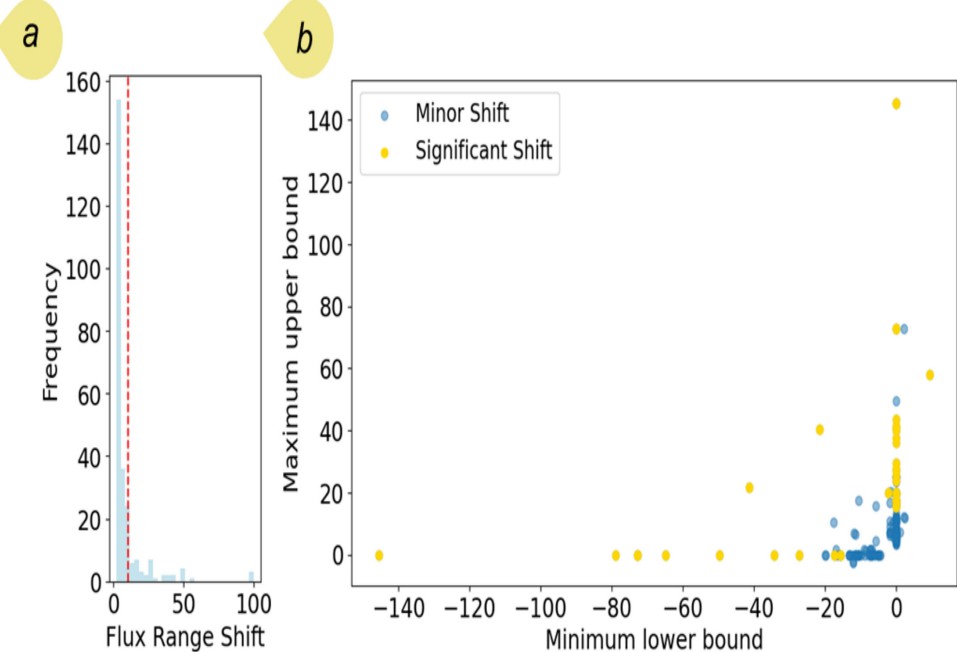

**FIG 8** (a) Histogram of maximal flux range shifts obtained from FVA simulations for *E. coli* bacteria. For a better representation, fluxes with a shift greater than 2 are shown in the figure (see Fig. S8 to see the entire range). The red line threshold used to define significant shifts is based on the 95th percentile for all fluxes; (b) minimum lower bound and maximum upper bound related to the fluxes in part (a). Fluxes are categorized with regard to the threshold.

that demonstrate substantial flux shifts, which could indicate critical metabolic roles or regulatory mechanisms (Fig. 8). For *E. coli* species, differences in flux ranges across the bacteria within the domain were significant (Wilcoxon signed-rank test, *P*-value < 0.001). Fig. 8A and B represent the *E. coli* reactions with flux range shifts greater than 2 mmol/g$_{DW}$/h (see the Methods section for the statistical analysis). While most reactions experienced minor shifts, some had significant shifts, with their absolute minimal lower bound and maximal upper bound exceeding 20. The enzymes with their associated GPRs involved in the reactions with the largest flux range shifts are listed in Table S13. From the results, acetaldehyde dehydrogenase, acetate kinase, acetolactate synthase, adenylate kinase, and adenylate kinase experienced the highest shifts.

## DISCUSSION

The quest to accurately model microbial ecosystems is at the heart of microbial ecology, biotechnology, and health sciences. As we delve deeper into the metabolic intricacies of microbial communities, the limitations of current models become more apparent, calling for approaches that better capture the underlying principles of microbial dynamics. By embracing advanced computational strategies and addressing the fundamental aspects of microbial growth and interactions, we stand on the cusp of advancing our ability to predict and harness the interplay of microbial communities for a myriad of applications.

Advancements in high-throughput omics technologies have been instrumental in determining microbiome composition and function (63). Although these technologies, primarily through fecal and colonic lavage sampling, have unveiled microbiota's spatial heterogeneity, robust computational methodologies are still needed to decipher underlying mechanisms. In this regard, multiscale modeling approaches, particularly AMBs, have shown promise. ABMs enable detailed prediction of microbial dynamics and spatial organization by representing individual microbial agents, their metabolic capabilities, and environmental interactions. Integrating ABM with GEMs further enhances this capability, linking detailed microbial metabolism to the broader ecological dynamics of their communities.

We simulated the dynamics and spatial pattern of MMCs at the proximal small intestine and the cecum. This simulation used physiologically relevant oxygen gradients, which reflected the conditions of these gastrointestinal regions. Our findings, based on the heterogeneity in metabolic fluxes among certain metabolites, showed that microbes, including *B. thetaiotaomicron*, could modulate their metabolic processes according to substrate availability. This flexibility allows *B. thetaiotaomicron* to efficiently adapt to local environmental conditions and optimize its metabolism to thrive in different niches within the gut. Liu et al. (64) described *B. thetaiotaomicron* as a metabolic generalist endowed with the ability to hydrolyze different polysaccharides. This functionality is particularly pivotal when alternative dietary nutrients are absent, which enables the bacteria to effectively settle in the mucus layer of the colon. Such colonization is important for its survival in the gut, as it adapts to using mucin as an alternative source when conventional dietary compounds are unavailable.

Our results can provide insights into the spatial regulation of metabolites, including amino acids, which yield significant benefits for therapeutic interventions. Different gastrointestinal tract regions show varying nutrient availability, oxygen levels, and even pH, affecting microbial communities and their metabolic interactions. Mapping these variations can help better understand how certain conditions might be impacted by localized microbial activities. This also facilitates the development of targeted therapies, such as localized drug delivery systems that act on specific microbial communities or their metabolism. This can be particularly useful in treating localized gut diseases (65), such as tumor-targeted therapy, without affecting the entire gut milieu. Research, for instance, has explored the use of oncolytic bacteria to regulate the immune system and suppress tumor progression by secreting tumor-killing nutrients, which improves colonization (66).

In this study, we specifically focused on microbial interactions occurring within the biofilm matrix, rather than attempting to simulate mucus-specific factors, such as mucin glycans or turnover rates. Given the current lack of targeted experimental data on microbial activities within MMCs, this targeted approach allows us to focus on the inherent complexity of these communities' dynamics. Though our framework did not encompass all the complexities associated with MMCs, it represents a step forward in modeling MMCs by integrating modeling approaches across different scales. This integration allowed us to address the limitations identified in earlier tools, particularly the lack of granularity in spatial and metabolic interactions. This integration supports the formulation of testable hypotheses aimed at uncovering core mechanisms governing MMCs. Furthermore, the ability to define specific boundary conditions for different metabolites enhances our model's capacity to simulate microbial communities close to the epithelium under varying environmental conditions.

A primary challenge in modeling MMCs in the gut is the scarcity of targeted experimental data. Commonly, experimental studies of gut ecosystems rely on analyses of fecal samples or gut lumen microbiota, which do not accurately represent MMCs close to the epithelial boundary and are notably challenging to sample. This limitation contributes to remarkable data gaps, affecting the precision of our model. As a result, we approximated our simulation results using available experimental datasets not explicitly tailored to distinct gut regions, and this methodological mismatch may explain some observed deviations. Additionally, the specialized environments within different gut regions require further refinement to improve simulation accuracy. On the contrary, our framework relies on GEMs, which are dependent on genome annotations that can be incomplete in some cases. While the biochemical pathways for *E. coli* are well-documented, allowing for detailed modeling of spatial regulation of amino acids such as alanine and glycine, this level of detail is not available for many other species. In addition, parameters in diffusion equations, such as diffusion coefficients, are important for predicting metabolite distribution and stability near the mucus boundary. Precise determination and validation of these parameters are essential to improve the model's predictive accuracy and reliability, as variations can lead to deviations in predictions.

Looking ahead, the inclusion of additional dietary components, such as fatty acids, represents a valuable direction for refining our model and expanding its applicability. High-protein diets often contain varying levels of dietary fats, which are metabolized differently from proteins and can impact microbial communities and SCFA production in the gut. Incorporating fatty acids into future simulations could yield insights into their potential effects on SCFA profiles and microbial interactions, particularly within the mucosal environment. This extension would allow us to capture more complex dietary impacts on microbial metabolic pathways, providing a more comprehensive view of diet-microbe interactions relevant to gut health.

The insights gleaned from our study not only shed light on the complex dynamics of microbial communities but also set the stage for future research to explore novel therapeutic strategies targeting microbial interactions at the mucosal barrier. Our findings underscore the importance of adopting a multiscale, multidimensional approach to accurately model the intricate ecosystem of microbial communities, thereby enhancing our understanding and management of microbial dynamics in health and disease.

## Conclusion

Modeling microbial ecosystems with precision is a fundamental pursuit in microbial ecology, biotechnology, and health sciences, as these systems influence a range of applications, from gut health to environmental stability. Traditional dynamic models have struggled with the complexity of microbial ecosystems, often limited by oversimplified representations of spatial and metabolic processes. *Meta*Biome represents a significant step forward in modeling MMC dynamics in spatially and environmentally complex contexts, particularly the human gut. By incorporating metabolite-mediated

interactions within a spatially heterogeneous framework, *Meta*Biome captures dynamics that are often simplified in traditional cell-centered models, adding layers of realism that reveal how microbial interactions are shaped by their surroundings.

Key features in *Meta*Biome, such as the shoving mechanism and boundary constraints, enable a more accurate representation of biofilm density and structure, including microbial organization along physical surfaces like the intestinal wall. These aspects allow for a more nuanced exploration of microbial communities in confined spaces, shedding light on how spatial structure and density influence community behavior. *Meta*Biome's environmental layer also employs FVM to resolve metabolite transport across gradients, providing insight into spatially varying metabolite distributions and their effects on microbial interactions.

With its computational efficiency achieved through a NumPy backend, *Meta*Biome supports high-frequency simulations, enabling researchers to conduct detailed explorations of MMCs under a variety of conditions. When applied to the gut microbiome, the framework allows for the study of potential cross-feeding interactions, prediction of biosynthesis of localized metabolites, and spatial regulation of metabolites, offering a glimpse into the metabolic organization and resilience of gut communities.

While our framework addresses specific challenges in modeling microbial communities, it also contributes to the broader effort to develop computational tools that capture the intricate dynamics of microbial ecosystems. By bridging the gap between metabolic modeling and spatial heterogeneity, it advances our understanding of microbial interactions in complex environments, offering a valuable resource for researchers aiming to predict and leverage the functional potential of microbiomes in health and environmental applications.

Future enhancements will focus on integrating additional layers of biological realism, such as the influence of host–microbe interactions and the dynamic physical forces that shape biofilm structure. These refinements aim to address existing gaps in our understanding of microbial ecology in the gut and enable broader applications of the model in investigating microbiome–host relationships and their implications for health and disease.

## MATERIALS AND METHODS

### Metabolic network reconstruction and refinement

Metabolic networks for different species are inferred from their genomic sequences. The genome sequences and GenBank genome files for species used in this study (Table S2) were downloaded from the National Center for Biotechnology Information. CarveMe was used to reconstruct the initial draft metabolic networks using the genome sequences (67). We used Mauve 2.4 to perform a homology search based on a comprehensive genome-wide multiple sequence alignment using the default parameter settings (68). The GenBank files were used to identify gene orthologous pairs (69). Using these homologous gene pairs, gene–protein–reaction (GPR) associations missing in the existing draft networks were determined and incorporated into the drafts. Next, to ensure the inclusion of all conceivable reactions, the models underwent a comparison in terms of reactions included with those provided in AGORA (70). In the next step, an in-house code interfacing with the Kyoto Encyclopedia of Genes and Genomes (KEGG) API was used to re-evaluate coding sequences lacking GPRs to incorporate the latest biochemical updates available on the KEGG database (71). All transport reactions were compared with relevant AGORA models. Different functionality tests were conducted. All GEMs underwent manual curation, drawing upon the details given in the literature. We conducted growth simulations on different carbon sources using the draft networks using data sources from existing literature when available to examine their capability (Table S3). We ensured that the models effectively synthesized the amino acids and SCFAs investigated in this study. The generated models were used for the main simulations (Table S4).

## Model description

Our study extends our previous efforts aimed at simulating mucosal microbial ecosystems in the gut by introducing a more complex, layered approach that integrates ABM, FVM, and GEMs. This multiscale model is designed to simulate microbial attachment, growth, and interaction dynamics of MMCs within the gut environment, capturing physical constraints on microbial populations and their chemical interactions through diffusion gradients and metabolite consumption and production. By simulating these processes, the model predicts the emergent spatial and temporal patterns and metabolic interactions of microbial communities.

### Model structure

Our analysis investigates a microenvironment close to the mucus layer, where microbes grow and are washed away into the lumen by shear forces at a defined height above the underlying epithelial surface (72–74). The simulation environment is modeled as a joint system in which individual bacterial agents within a two-dimensional (2D) domain interact with metabolites modeled using the FVM. We ensured that all simulations reached a steady state, indicating no significant fluctuations in the system's state variables, including metabolite concentrations and microbial abundances.

### Agent layer

At a high level, the ABM for bacterial populations models individual bacteria as discrete agents on a 2D grid. Each bacterium is assumed to be spherical and characterized by parameters representing mass, radius, position, and species type. They are embedded in a finite volume-based continuum model of metabolic concentrations in order to allow for feeding and growth behaviors. Biofilm dynamics are modeled by updating bacteria growth, feeding, and division in discrete timesteps. We implement the ABM package in Python 3.10, using the NumPy (75) and SciPy (76) packages for numerical operations.

### Bacterial agents

The agent layer models individual microbial agents, capturing their positional information and dynamic behaviors, such as movement, growth, division, and mechanical interactions. Mechanical interactions among agents are handled using a shoving algorithm to prevent overlapping and maintain spatial separation. In our model, each microbe is represented as a spherical agent governed by specific rules for cellular processes. Agents grow and divide based on local nutrient availability, with growth rates defined by the metabolic pathway layer. Then, at each time step, the shoving algorithm is updated to push bacteria apart when their volumes intersect. Collision constraints ensure that agents do not intersect with substratum/side boundaries or each other.

### Collision constraints

The ABM simulation uses a shoving algorithm to prevent bacteria from intersecting with each other or with boundaries within the simulation domain. For each pair of bacteria, the displacement distance is defined as the center-to-center distance subtracted from the sum of their radii. Bacteria are intersecting if their displacement distance is positive. During a single shoving iteration, for each bacterium, all intersecting neighbors are calculated. For each neighbor, a vector with a length equal to half of the displacement distance directed along the line toward the neighbor's center is calculated. This vector is added to a cumulative shoving vector, representing the total pushing from all neighbors. The candidate update position of the bacterium is then determined by its original position added to the shoving vector.

To complete a shoving iteration, bacteria are moved along the line from their initial to candidate positions until they would intersect a wall within the simulation domain. Their final position is set to be this point. Since a single shoving iteration may not

resolve all intersections, the shoving and wall collision steps are iteratively applied until convergence is achieved, typically within roughly eight iterations.

### Model geometry

The model geometry is a 2D box divided into lattice cells measuring 10 μm × 10 μm (see Table S14 for the domain size details). The bottom boundary is impermeable to bacteria, while the side boundaries are assigned free boundary conditions with a wrapped environment assumption. The biofilm is modeled with a maximum height as specified in Table S14; any bacteria exceeding this height are considered sloughed off and are subsequently removed from the simulation.

### Environment layer

We assume that solute (metabolite) concentrations are approximately constant within each lattice cell and model these concentrations through partial differential equations solved using the FVM. All concentration gradients are modeled using the Python FiPy package (77).

Nutrients enter the model domain from the upper boundary at fixed concentrations (Dirichlet condition) and diffuse across the domain. Metabolites produced during bacterial growth are set to zero concentration at the top boundary to simulate diffusion into the lumen, while extracellular metabolites are maintained under a no-flux condition (Neumann condition) at the bottom boundary. In certain scenarios, specific metabolites, including SCFAs, are absorbed to mimic the uptake by host cells, which serve as a sink for these compounds. In the high-protein diet scenario, we applied a vertical oxygen gradient to replicate physiological oxygen conditions close to the mucus layer, which was presumed to originate from the highly vascularized subepithelial mucosa (47) (Table S5).

For each species in each lattice cell, we estimate density as the sum of mass and treat bacteria density as a continuous property for the purpose of the FVM layer. Under the assumption that solutes diffuse on timescales much faster than bacterial growth, solute concentrations are then calculated as the steady-state solutions of reaction–diffusion equations. In this model, bacteria act as sources and sinks for metabolites based on their metabolic activity, as determined by GEMs, and metabolites are allowed to diffuse between cells according to their diffusion coefficients. The overall system satisfies the following equation:

$$\nabla \cdot \left( D_{i,m} \cdot \nabla S_i \right) + R_i = 0,$$

where $D_{i,m}$ is the diffusion coefficient of metabolite $i$ in medium $m$, either biofilm or bulk, and $S_i$ is the metabolite concentration for metabolite $i$. Diffusion coefficients were adjusted for the biofilm matrix typically set to 0.6 times the diffusion in the bulk fluid (78–80) (Table S12). $R_i$ represents the net local consumption or production rate directly obtained from the metabolic fluxes of agents in each lattice cell calculated by the metabolic pathway layer.

### Metabolic pathway layer

The dynamic metabolic interactions between agents and their environment are crucial for capturing the emergent properties of MMCs. Each agent's metabolism influences its growth and division rates, contributing to the overall spatial and temporal patterns observed in the simulation. Integrating GEMs into the ABM, each agent is endowed with a GEM that dictates its metabolic fluxes based on the local conditions simulated in the environment layer. We employed constraint-based models to optimize intracellular fluxes, which determine substrate consumption and product formation rates using the COBRApy toolbox (81). These models calculate intracellular metabolic fluxes that drive the agent's growth and metabolic outputs, impacting local concentrations of substrates and products in the gut environment.

At each time step, using the metabolite concentrations from the environment layer, lower fluxes of relevant uptake reactions were constrained based on Michaelis–Menten kinetics (82) (listed in Table S12). Following an FBA-based method, flux distributions were obtained to update agents' biomass and metabolite concentrations.

### Flux balance analysis

FBA was used to optimize the flux of reactions in the metabolic networks. FBA assumes no accumulation of metabolites occurs during growth, while the objective function, in our case, the biomass-producing reaction, is maximized (83). The biomass-producing reaction is a weighted ratio of components in cell dry weight maximized based on the evolutionary assumption (84). In the present study, biomass-producing reactions introduced by AGORA (70) were integrated into our reconstructed models and set as the objective function.

### Parsimonious flux balance analysis

$p$FBA is an extension to FBA, which aims at limiting the total flux throughout the whole network by imposing an $l_1$-norm constraint while still optimizing the objective function (85).

### Flux variability analysis

FVA is an optimization method used to explore the range of possible fluxes through reactions in the metabolic network, providing insights into its flexibility and robustness (86). For the simulations, we bound the objective function to 90% of its maximal values obtained by FBA.

## Simulation time loop

The overall simulation proceeds in discrete timesteps, each of which is sequentially divided into movement, solute equilibrium, and growth and division steps.

- In the movement phase, agents outside the biofilm are moved by 5 µm along a direction vector, and their direction vector is randomly rotated by a normally distributed number of degrees. This simulates bacterial diffusion in the lumen, while agents within the biofilm are solely driven by mechanical pushing from division and growth. Collisions with walls or other bacteria are resolved through iterative adjustments of shoving and wall constraint adjustments limited to 10 repetitions to maintain computational efficiency.
- During the solute equilibrium phase, solute concentrations are calculated according to the boundary conditions and metabolic activity levels described previously. For each bacterial species within each lattice cell, growth rates are extracted from the mass growth term in the corresponding feeding equation.
- During the growth and division phase, bacteria increase in mass according to growth rates derived from biomass fluxes in the metabolic pathway layer constrained by the metabolite availability and the enzymatic capacity of the cells. This increase leads to an update in their radii. Bacteria exceeding a mass of 2 µm are split into two equally sized offspring, each randomly positioned adjacent to their parent's center.

## Initialization and implementation

Simulations start with randomly placed agents within the domain, simulating the initial attachment phase where microbes are free to move throughout the domain until they adhere to the substratum. Each bacterial population starts with an initial size and density. Nutrients enter from the upper boundary with a given concentration and diffuse throughout the domain to be consumed by the bacteria. The system evolves over time, with bacterial populations performing metabolism, growing, and dividing, while

metabolites are added, removed, and diffused throughout the grid. The nutrients enter from the domain with predetermined concentrations specific to each nutrient. Unless stated otherwise, we presumed that fibers and carbohydrates were converted to glucose to the same extent. Details about media, initial conditions, the constraint-based model, and metabolites enabled in the environment layer for the scenarios are provided in Table S5 to S9.

The model includes a range of environmental conditions to simulate different scenarios. These conditions include the presence of oxygen, varying levels of nutrients, and different initial compositions of the microbial community. Output from the model includes the spatial and temporal distributions of bacterial populations and metabolite concentrations. These data are analyzed to understand the development of microbial communities, their metabolic interactions, and their response to environmental changes. Key metrics include the growth rates of bacterial populations, the production and consumption of key metabolites, and the spatial distribution of microbial diversity.

## Statistical analysis

FVA simulations were conducted on all agents within the domain, employing optimized constraints derived from preliminary simulations. This optimization ensured that the metabolic models were tailored to reflect environmental conditions. The ranges of fluxes obtained from FVA were subjected to statistical comparison using the Wilcoxon signed-rank test. This non-parametric test was chosen due to its robustness in comparing paired data, which, in this case, involved the FVA results of agents at different locations in the domain. The resulting $P$-values provided a measure of the statistical significance of the differences observed. We generated a histogram for all reactions specific to a species to characterize the shifts in flux ranges. The 95th percentile was used to establish a threshold for significant flux shifts. Reactions with the highest range shift were mapped to their corresponding gene–protein associations, thereby pinpointing critical metabolic reactions and their associated genes that could be targeted to alter microbial metabolic behavior.

## ACKNOWLEDGMENTS

This work was partially supported by funding from the University of California Berkeley and by National Science Foundation (Award # 1728407 to M.R.K.M.). A.V. was partially supported by Natural Sciences and Engineering Research Council of Canada (NSERC Postdoctoral Fellowship) and Fonds de Recherche du Québec (FRQNT Postdoctoral Fellowship). Publication was made possible in part by support from the Berkeley Research Impact Initiative (BRII) sponsored by the UC Berkeley Library.

## AUTHOR AFFILIATIONS

[1]Molecular Cell Biomechanics Laboratory, Departments of Bioengineering and Mechanical Engineering, University of California, Berkeley, California, USA
[2]Molecular Biophysics and Integrative Bioimaging Division, Lawrence Berkeley National Lab, Berkeley, California, USA

## AUTHOR ORCIDs

Javad Aminian-Dehkordi  http://orcid.org/0000-0002-4936-6535
Andrew Dickson  http://orcid.org/0000-0002-1146-6346
Mohammad R. K. Mofrad  http://orcid.org/0000-0001-7004-4859

## AUTHOR CONTRIBUTIONS

Javad Aminian-Dehkordi, Conceptualization, Data curation, Formal analysis, Investigation, Methodology, Software, Visualization, Writing – review and editing | Andrew Dickson, Resources, Software, Visualization, Writing – review and editing | Amin Valiei,

Conceptualization, Investigation, Visualization, Writing – review and editing | Moham-mad R. K. Mofrad, Conceptualization, Funding acquisition, Investigation, Methodology, Project administration, Supervision, Validation, Visualization, Writing – review and editing

## DATA AVAILABILITY

*Meta*Biome is implemented in Python and from GitHub (https://github.com/mofra-dlab/MetaBiome).

## ADDITIONAL FILES

The following material is available online.

### Supplemental Material

**Supplemental Information (mSystems01652-24-S0001.pdf).** Supplemental figures and tables.

### Open Peer Review

**PEER REVIEW HISTORY (review-history.pdf).** An accounting of the reviewer comments and feedback.

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
