## [Reviewer comments · mSystems]

***Meta*Biome: A Multiscale Model Integrating Agent-Based and Metabolic Networks to Reveal Spatial Regulation in Gut Mucosal Microbial Communities**

Javad Aminian-Dehkordi, Andrew Dickson, Amin Valiei, and Mohammad Mofrad

Corresponding Author(s): Mohammad Mofrad, University of California Berkeley

Review Timeline:

Submission Date:	December 9, 2024
Editorial Decision:	January 7, 2025
Revision Received:	January 28, 2025
Accepted:	March 4, 2025

Editor: Samuel Chaffron

Reviewer(s): The reviewers have opted to remain anonymous.

Transaction Report:

DOI: <https://doi.org/10.1128/msystems.01652-24>

Re: mSystems01652-24 (MetaBiome: A Multiscale Model Integrating Agent-Based Modeling and Metabolic Networks Reveals Spatial Regulation in Mucosal Microbial Communities)

Dear Prof. Mohammad Mofrad:

As pointed out by reviewer #1, prior to acceptance for publication, it is essential that the code and data should be made freely available at the GitHub page that has been setup: <https://github.com/mofradlab/MetaBiome>
Also I strongly recommend you to address that latest comments of reviewer #2.

Revision Guidelines

Sincerely,
Samuel Chaffron
Editor
mSystems

Reviewer #1 (Comments for the Author):

I commend the authors addressing my comments and clarifying the properties of the modeling framework and comparing with existing tools. The novelty of the work now becomes more clear.

I have the following remaining comments:

1. I appreciate that a GitHub link for the code was added on page 39, but unfortunately, the link does not work.
2. If the modeling framework is a work in progress as mentioned in the rebuttal, then a paragraph on future directions should be added in the Conclusion.

Reviewer #2 (Comments for the Author):

In the revised manuscript, the authors have sufficiently addressed major questions raised in the initial submission. The revised text and figures also effectively extend these clarifying details to the main manuscript. The added text expanding upon the computational framework of MetaBiome also helps underline this platform's advancement to multi-scale representations of gut microbiomes. Although not revolving around MMC systems, a recent study of interest (DOI: 10.1371/journal.pcbi.1012031) may further bolster the Introduction's motivation towards a species interaction-focused, spatially-resolved extension of tools like BacArena, COMETS, and ACBM for simulating microbial communities.

The reviewer thanks the authors for performing the supplemental simulations and analysis on *E. coli* neighbor diversity in order to evaluate its influence on metabolic exchange flux outputs. Although this *E. coli* analysis is sufficient for this manuscript's context, could this non-significant difference be somewhat expected in *E. coli* as its cross-feeding components can originate from the majority consortium member or all members in the case of aldehyde? It would be interesting to conduct similar simulation analysis when select community members are heavily reliant on a specific cross-feeding relationship from a non-majority species.

The added MetaBiome simulation details, specifically those between Page 36 and 37 elaborating the agent "cell motility", raised a new set of questions around agent-level behaviors in the MMC. Simulation output analysis approximated the biofilm area having a maximum bulk thickness of 80 μm but Figure 4B appears to suggest that cell agents can still propagate/swim above this biofilm height after being sloughed off? From these observations, can cell agents that entered this intermediary region re-adhere to the biofilm surface (if it can locate a grid opening below the 80 μm mark)?

A detail that the reviewer neglected to emphasize during the first review was the presentation of the simulated gut environment in Figure 7. With text in the figure referring to putrescine and ornithine metabolic pathways, one can infer that Figure 7A is of the proximal small intestine from what was discussed in earlier result sections, but a quick clarifying label either in the figure or caption would go a long way. Additionally, a supplemental figure duplicating Figure 7B but for the cecum would provide a quick display of the "efficient conversion of putrescine to other compounds" stated in Page 20.

Dear Prof. Mohammad Mofrad:

As pointed out by reviewer #1, prior to acceptance for publication, it is essential that the code and data should be made freely available at the GitHub page that has been setup:

<https://github.com/mofradlab/MetaBiome>

Also I strongly recommend you to address that latest comments of reviewer #2.

Revision Guidelines

To submit your modified manuscript, log into the submission site at

<https://msystems.msubmit.net/cgi-bin/main.plex>. Go to Author Tasks and click the appropriate manuscript title to begin. The information you entered when you first submitted the paper will be displayed; update this as necessary. Note the following requirements:

Responses to Reviewer #1

1. Reviewer: I commend the authors addressing my comments and clarifying the properties of the modeling framework and comparing with existing tools. The novelty of the work now becomes more clear. I have the following remaining comments:

Our Reply:

We thank the reviewer for their kind acknowledgment of the revisions made. Your feedback has been instrumental in enhancing the quality and clarity of our manuscript, and we greatly appreciate your review.

2. Reviewer: I appreciate that a GitHub link for the code was added on page 39, but unfortunately, the link does not work.

Our Reply:

The GitHub link is now available for public use (<https://github.com/mofradlab/MetaBiome>).

3. Reviewer: If the modeling framework is a work in progress as mentioned in the rebuttal, then a paragraph on future directions should be added in the Conclusion.

Our Reply:

In response, we have added a concise discussion outlining the planned advancements for our modeling framework. These include incorporating additional biological and physical features to improve its applicability in studying gut mucosal microbial dynamics. Page 28 now reads:

“Future enhancements will focus on integrating additional layers of biological realism, such as the influence of host-microbe interactions and the dynamic physical forces that shape biofilm structure. These refinements aim to address existing gaps in our understanding of microbial ecology in the gut and enable broader applications of the model in investigating microbiome-host relationships and their implications for health and disease.”

Responses to Reviewer #2

3. Reviewer: In the revised manuscript, the authors have sufficiently addressed major questions raised in the initial submission. The revised text and figures also effectively extend these clarifying details to the main manuscript. The added text expanding upon the computational framework of MetaBiome also helps underline this platform's advancement to multi-scale representations of gut microbiomes. Although not revolving around MMC systems, a recent study of interest (DOI: 10.1371/journal.pcbi.1012031) may further bolster the Introduction's motivation towards a species interaction-focused, spatially-resolved extension of tools like BacArena, COMETS, and ACBM for simulating microbial communities.

Our Reply:

We sincerely thank the reviewer for acknowledging the revisions made. We are pleased that the added text and figures have effectively clarified and expanded upon our model. We also appreciate the suggestion to include the MiMICS framework as part of the Introduction. This work provides valuable insights into the importance of integrating spatial transcriptomics with multiscale modeling to understand complex metabolic behaviors.

Paragraph 2, page 5 now reads:

“Previous ABM-based models integrated with metabolic networks have significantly advanced our understanding of microbial communities in general, yet they have generally lacked a specific focus on interactions within MMCs and the underlying mechanical behavior between species. Most of these previous models primarily focused on population-level interactions, often without the detailed spatiotemporal resolution required to effectively study biofilm structure. For instance, BacArena (33), an R package based on MatNet (34), adopts an ABM approach that places each cell on a grid block, which represents microbial communities in a spatial layout. While this setup allows for species interactions and spatial positioning, BacArena is limited by its cell-as-grid-block design. It fails to capture important biological phenomena like agent cell size, movement dynamics, or intracellular variability, and it restricts the ability to model dynamic changes in transport properties and diffusion mechanisms with high fidelity. COMETS, implemented in Java with MATLAB and Python toolboxes, goes a step further by leveraging dynamic FBA and adopts a population-centric approach over ABM, focusing solely on calculating the average metabolic activities of cell populations within each grid unit (35). As a result, it is less suited for studying heterogeneous environments, particularly MMCs, where local nutrient gradients and cell-level interactions are crucial. Versluis et al. (36) also used a similar approach to study microbial transitions in the infant gut microbiota., which, while informative, lacked the spatial resolution needed to address MMC-specific dynamics. Recently, MiMICS has developed as multiscale model using ABM and incorporating spatially resolved transcriptomic data to identify distinct metabolic states within *Pseudomonas aeruginosa* biofilms, providing insights into the development of targeted strategies to manage biofilm associated infections (37).”

4. Reviewer: The reviewer thanks the authors for performing the supplemental simulations and analysis on *E. coli* neighbor diversity in order to evaluate its influence on metabolic exchange flux outputs. Although this *E. coli* analysis is sufficient for this manuscript's context, could this

non-significant difference be somewhat expected in *E. coli* as its cross-feeding components can originate from the majority consortium member or all members in the case of aldehyde? It would be interesting to conduct similar simulation analysis when select community members are heavily reliant on a specific cross-feeding relationship from a non-majority species.

Our Reply:

To address the reviewer's question, initially, when we were trying to address the reviewer's previous comment on *E. coli*, we extended the analysis to other community members within the simulation, using a similar methodology as with *E. coli*. However, we were unable to identify any statistically robust, meaningful patterns in metabolic flux differences within the domain. We realized that the rich medium used in our simulations provided multiple nutrient sources, enabling microbial growth and metabolite production to rely on a variety of pathways. This complexity likely masked the effects of neighbor diversity and made it challenging to isolate specific dependencies or patterns using traditional analysis methods.

We found this question highly stimulating and believe it highlights an important area for further investigation. Our analysis suggests that the interplay of multiple variables, such as medium components and species-specific metabolic behaviors, complicates efforts to attribute exchange flux differences to neighbor diversity using conventional statistical approaches. So we have identified this as an area where advanced analytical tools may provide deeper insights.

To this end, we have initiated a new direction in our ongoing research. We are currently exploring the use of advanced network analysis techniques, complemented by machine learning, to study the complex interactions between microbial community members. This approach will allow us to systematically study the effects of individual medium components and better quantify the influence of specific cross-feeding relationships. While this work is still in progress, it represents an exciting step forward in addressing the complexities of microbial interactions in the gut.

5. Reviewer: The added MetaBiome simulation details, specifically those between Page 36 and 37 elaborating the agent "cell motility", raised a new set of questions around agent-level behaviors in the MMC. Simulation output analysis approximated the biofilm area having a maximum bulk thickness of 80 μm but Figure 4B appears to suggest that cell agents can still propagate/swim above this biofilm height after being sloughed off? From these observations, can cell agents that entered this intermediary region re-adhere to the biofilm surface (if it can locate a grid opening below the 80 μm mark)?

Our Reply:

We appreciate the reviewer highlighting this point, as it prompted us to enhance the clarity of the manuscript. To clarify, the biofilm thickness of 80 μm mentioned in the manuscript relates specifically to the high-protein diet scenario. Scenarios I and II were performed with a biofilm thickness of 100 μm . Once detached, bacteria are not capable of re-adhering, as they are effectively removed from the simulation.

As indicated in the SI file (Table S14), the domain width for these scenarios is listed as 120 μm , while it is 100 μm for the high-protein diet scenario. To avoid confusion, we have added a new column to Table S14 specifying the biofilm thickness for each scenario. The "Model geometry" section, page 32 now reads:

"The biofilm is modeled with a maximum height as specified in Table S14; any bacteria exceeding this height are considered sloughed off and are subsequently removed from the simulation."

Table S14- Size of grids used for different experiments.

Scenario	length (μm)	Width (μm)	Lattice site (μm)	Biofilm length (μm)
Scenario 1	400	120	10	100
Scenario 2	400	120	10	100
High protein-diet on overweight individuals	800	100	10	80

6. Reviewer: A detail that the reviewer neglected to emphasize during the first review was the presentation of the simulated gut environment in Figure 7. With text in the figure referring to putrescine and ornithine metabolic pathways, one can infer that Figure 7A is of the proximal small intestine from what was discussed in earlier result sections, but a quick clarifying label either in the figure or caption would go a long way. Additionally, a supplemental figure duplicating Figure 7B but for the cecum would provide a quick display of the "efficient conversion of putrescine to other compounds" stated in Page 20.

Our Reply:

We thank the reviewer for their insightful feedback regarding Figure 7. we have made the following revisions:

1. The caption for Figure 7 is updated to explicitly point out that it represents the proximal small intestine. Figure 7's caption, page 21 now reads:

“Figure 7- Results of spatial regulation of reaction fluxes for *E. coli* in the proximal small intestine scenario, (a) Simulated environment showing the spatial position of two *E. coli* bacteria: one located near the top boundary with lower access to oxygen but higher access to lumen derived metabolites (position A), and the other, near the bottom boundary, closer to the host surface (position B). (b) Suboptimal FVA simulations for *E. coli*. Red and green values correspond to agents at positions A and B, respectively, highlighting the differences in flux patterns due to spatial positioning. To see the differences in fluxes across all *E. coli* bacteria in the cecum scenario, refer to Figure S8. ”

2. Regarding the reviewer's comment, we generated a supplemental figure analogous to Figure 7B with a focus on the cecum scenario, providing additional support for our findings.

Figure S8- Suboptimal FVA simulations across all *E. coli* agents in the cecum scenario. Values in red correspond to all possible fluxes. FVA simulations for all *E. coli* bacteria in the domain were performed and the union of flux intervals was reported (related to Figure 7).

Re: mSystems01652-24R1 (*MetaBiome: A Multiscale Model Integrating Agent-Based and Metabolic Networks to Reveal Spatial Regulation in Gut Mucosal Microbial Communities*)

Dear Prof. Mohammad Mofrad,

Your manuscript has been accepted, and I am forwarding it to the ASM production staff for publication. But please do note the latest comment by Reviewer 1. Your paper will first be checked to make sure all elements meet the technical requirements. ASM staff will contact you if anything needs to be revised before copyediting and production can begin. Otherwise, you will be notified when your proofs are ready to be viewed.

Cover Image Submissions: If you would like to submit a potential Cover Image, please email a file and a short legend to mssystems@asmusa.org. Please note that we can only consider images that (i) the authors created or own and (ii) have not been previously published. By submitting, you agree that the image can be used under the same terms as the published article. Image File requirements: TIF/EPS, 7.5 inches wide by 8.25 inches tall (at least 2,250 pixels wide by 2,475 pixels tall), minimum 300 dpi resolution (600 dpi preferred), RGB, and no figure elements, e.g., arrows or panel labels. The legend should be a short description of the image, 1-2 sentences recommended. Please download and use this interactive template in Adobe to ensure that your proposed cover image meets our size requirements (<https://journals.asm.org/pb-assets/pdf-text-excel-files/ASM-Interactive-Sizing-Cover-Template-1715689791.pdf>).

We recognize that the video files can become quite large, so to avoid quality loss ASM suggests sending the video file via <https://www.wetransfer.com/>. When you have a final version of the video and the still ready to share, please send it to mSystems staff at mssystems@asmusa.org.

Sincerely,

Samuel Chaffron
Editor
mSystems

Reviewer #1 (Comments for the Author):

Please note that a reference is missing in line 124. Other than that, I have no further comments.

Reviewer #2 (Comments for the Author):

The authors did well addressing critiques.